# Immunomodulation by 4-Hydroxy-TEMPO (TEMPOL) and Dimethyl Fumarate (DMF) After Ventral Root Crush (VRC) in C57BL/6J Mice: A Flow Cytometry Analysis

**DOI:** 10.3390/biology14050473

**Published:** 2025-04-25

**Authors:** Maria Fernanda Vannucci Balzani, Lilian de Oliveira Coser, Alexandre Leite Rodrigues de Oliveira

**Affiliations:** 1Department of Structural and Functional Biology, Laboratory of Nerve Regeneration, Institute of Biology, University of Campinas (UNICAMP), Campinas 13083-862, SP, Brazil; mafevannucci@gmail.com (M.F.V.B.); coser.lilian@gmail.com (L.d.O.C.); 2Center for Gender-Specific Biology and Medicine (CGBM)–Brazil, University of Campinas (UNICAMP), Campinas 13083-862, SP, Brazil

**Keywords:** flow cytometry, ventral root lesion, immunoinflammation

## Abstract

Spinal nerve root lesions can occur after trauma or pathological processes, leading to dysfunction of brachial and lumbosacral plexuses and, consequently, sensory and motor losses. In the present study we evaluated drugs with immunomodulatory and neuroprotective potential with the aim of control the inflammatory response caused by nerve root injury. Treatment with TEMPOL and DMF effectively reduced polarization of glia, macrophages and lymphocytes towards pro-inflammatory profiles. However, TEMPOL action was more intense up to the second week post injury while DMF have shown positive effects at the third week post injury, indicating a complementarity of the combination of such drugs.

## 1. Introduction

Spinal nerve root injuries can occur due to motor vehicle accidents, firearm injuries, and falls, resulting in trauma to the brachial and lumbosacral plexuses. These injuries can either involve avulsion or compression, leading to the retrograde death of motoneurons and loss of synaptic inputs, if the ventral roots are affected [1]. Crushing injuries to nerve roots are commonly present in combination with herniated discs, spinal stenosis, and tumors, causing impairment of motor responses including reflexes. In this scenario, neuronal loss is also present, but it is less severe compared to avulsion injuries. Thus, while the loss in avulsion injuries is about 80%, in compressive injuries it is approximately 50%, according to established experimental models with rodents [2].

The mechanical rupture or stretching of the rootlets generates a hemorrhagic area in the interface of the white and gray matter of necrosis. Subsequently, an endogenous secondary injury occurs, marked by increased permeability of the blood-brain barrier, apoptosis of glial cells and neurons, and a complex neuroinflammatory response that persists for months or years after the initial trauma [3]. During the secondary injury development, there is also activation of the innate and adaptive pathways of the immune response. Activated microglia differentiate into a pro-inflammatory profile, secreting neurotoxic and pro-inflammatory agents, such as cytokines. For example, tumor necrosis factor (TNF-α) and interleukin-6 (IL-6) are secreted, activating the iNOS enzyme together with the nuclear factor κB (NFκB) signaling pathway and consequently promoting the synthesis of free radicals [4]. The final stages of the secondary injury, which can extend significantly overtime, are marked by the formation of cysts and cavitation and the establishment of an astroglial scar [5]. One of the most notable transformations is Wallerian degeneration, which occurs distal to the injury, where there is axonal fragmentation and phagocytosis of cellular debris, as well as preparation of the microenvironment for the regenerative process [2,6,7].

Astrocytes play an important role in removing the neurotransmitter glutamate from the extracellular space. However, after injury, this function is inhibited by TNF-α, IL-1β, and reactive oxygen species (ROS), leading to an increased concentration of glutamate in the extracellular environment [2,8]. Consequently, excitotoxicity occurs, a neurodegenerative process initiated by excessive activation of receptors for this neurotransmitter, leading to intracellular alterations such as the high permeability to Ca^2+^ and the production of reactive oxygen and nitrogen species, contributing to secondary cell death. Of note, the ROS increases even more because, after the injury, motoneurons buildup metabolism for survival and regeneration [8]. Excitotoxicity, among other factors, contributes to the spread of neuronal tissue damage to other regions, both rostrally and caudally to the lesion epicenter. This can result in secondary damage, leading to greater neuronal loss, oligodendrocyte death, and demyelination [9].

Considering the above, immunomodulatory and antioxidant drugs have the potential to minimize and prevent damage caused by ROS, as they can promote a pro-regenerative, anti-inflammatory profile. This induces the release of neuroprotective substances such as insulin-like growth factor 1 (IGF-1), nerve growth factor (NGF) and brain-derived neurotrophic factor (BDNF). Molecules that double as neuroprotective and immunomodulators have been studied in our laboratory. In particular, TEMPOL (4-hydroxy-2,2,6,6-tetramethylpiperidine-1-oxyl) and Dimethyl Fumarate (DMF) (N-N-Dimethylformamide have shown positive results following nerve root injuries [1,2,10,11].

TEMPOL is a low molecular weight molecule and can be considered a mimetic of superoxide dismutase (SOD), as it can catalytically react with superoxide [12,13]. It also has properties that allow it to cross the blood-brain barrier to capture ROS radicals. Thus, it has been shown to be effective as a neuroprotective agent in studies of ocular neurodegenerative diseases and ALS [14,15]. The neuroprotective effect of this drug has also been observed in a Parkinson’s disease study, as it induces the production of growth factors in differentiated PC12 cells, thereby reducing the effects of superoxide radicals and neuronal death [16]. Spejo and colleagues (2019) demonstrated that TEMPOL has anti-inflammatory effects, reducing astroglial and microglial reactions and preserving synapses after a motor root crushing injury in rats [2].

In our laboratory, TEMPOL has been evaluated alone and combined with DMF (NeuroBoost). DMF is a fumaric acid ester, administered orally and rapidly cleaved into monomethyl fumarate (MMF) [11,17,18,19]. Studies have shown that DMF has immunomodulatory, anti-inflammatory, and antioxidant effects, demonstrating significant efficacy in the treatment of MS, together with preclinical evidence of positive effects on ALS [20]. In a study of sciatic nerve crush in mice, DMF demonstrated anti-inflammatory modulation of the immune response, contributing to axonal regeneration and motor recovery [11]. The immunological impact of this drug is based on the inhibition of pro-inflammatory cytokine production, such as IFN-γ, secreted by Th1 lymphocytes, while stimulating the generation of cytokines with a Th2 profile, including IL-4 and IL-5 [21]. Another study highlighted that the administration of DMF after nerve root crushing induced a neuroprotective effect by redirecting the immune response towards an anti-inflammatory profile and preserving most of the synaptic connections of the injured motor neurons [1]. Thus, the aim of the present study was to analyze, by using flow cytometry, the immunomodulatory effects of Tempol and DMF in response to ventral root crush injury in mice, evaluating the subpopulations of astrocytes, microglia, macrophages, and lymphocytes.

## 2. Materials and Methods

This is a preclinical study designed to evaluate the effects of pharmaceutical treatments (TEMPOL and DMF) on immune and cellular responses following spinal cord injury in mice. The study employs flow cytometry for detailed cellular analysis and statistical analysis to determine the significance of the findings.

### 2.1. Animals

Female C57BL/6J-Unib mice, aged 6–8 weeks, were acquired from the Multidisciplinary Center for Biological Investigation at the University of Campinas (CEMIB/UNICAMP) and housed in the animal facility of the Laboratory of Nerve Regeneration, located at the Institute of Biology, UNICAMP. The animals were maintained under controlled conditions, including a 12-h light/dark cycle, regulated temperature and humidity, and *ad libitum* access to water and pelleted food. The experiments were conducted with approval from the Institutional Committee for Ethics in Animal Experimentation (Committee for Ethics in Animal Use—Institute of Biology—CEUA/IB/UNICAMP, protocols 5773-1/2021 and 6329-1/2023) and complied with the guidelines established by the Brazilian College for Animal Experimentation. Altogether, 45 mice were used, and the experimental groups are detailed in the Figure 1.

### 2.2. Surgical Procedure and Drug Treatment

Mice were anesthetized with intraperitoneal injections of ketamine (100 mg/kg, Sespo Ind. e Com., Paulínia/SP-Brazil) and xylazine (20 mg/kg, Syntec do Brasil Ltda., Santana do Parnaíba/SP-Brazil), and the fur on the left thigh was shaved. They were then placed on a heating pad set at 37°C, and anesthesia was maintained using isoflurane (1–2%, inhaled) to achieve a surgical plane. The spinal erector muscles were removed, and a hemilaminectomy of the L1 and L2 vertebrae was carried out to expose the nerve roots connected to the lumbar intumescence. The ventral roots at the L4, L5, and L6 spinal segments on the right side of the spinal cord were crushed using fine forceps (No. 4) for 10 s each, twice. After the surgery, the roots and muscles were repositioned, and the skin was sutured. The animals were placed in a warm environment for recovery from anesthesia and were given tramadol hydrochloride (5 mg/kg, orally) once daily for three consecutive days as post-operative analgesia.

The treatments (methylcellulose (vehicle), TEMPOL—20 mg/kg, and DMF—15 mg/kg) were administered daily (7, 14 or 28 dpi) by gavage starting one hour after analgesia. TEMPOL and DMF were diluted in methylcellulose.

### 2.3. Animals’ Euthanasia

Ventral root crushed mice that were treated with vehicle, TEMPOL or DMF were euthanized at 7, 14 or 28 days following injury (*n* = 5/treatment/day; Table 1). Mice were deeply anesthetized (300 mg/kg ketamine and 30 mg/kg xylazine) and transcardially perfused with buffered saline (NaCl 0.9% in sodium phosphate buffer—P.B 0.1 M, pH 7.38).

### 2.4. Flow Cytometry

Based on the protocol described by Coser and colleagues [22], following euthanasia, the spinal cords were dissected out under a surgical microscope (DFV Vasconcelos, Brazil), and the ipsilateral side of the lumbar intumescence was removed. To obtain a suspension of cells, the tissue was mechanically dissociated by passing through two meshes of 140 µm and 70 µm. The cell suspension was transferred to centrifuge tubes containing different concentrations of Percoll (Sigma, St. Louis, MO, USA): 70%, 50%, 37%, and 10%. The tubes were centrifuged at 400× *g* for 30 min at 4 °C (acceleration 8, deceleration 0, and no change in bucket) for cell isolation. Afterward, the 10/37, 37/50, and 50/70 fractions were stimulated for 3 h at 37 °C in a CO_2_ incubator (5%). The stimuli used were PMA (5 µL, 50 ng/mL), Ionomycin (5 µL, 250 ng/mL), and Brefeldin A (5 µL, 1 µg/mL), all diluted in 1 mL of DMEM supplemented with 10% FBS and 1% penicillin/streptomycin (1 mg/1 g, Vitrocell, Waldkirch, Germany).

Subsequently, the cells were washed and fixed with commercial buffers (kit True-Nuclear Transcription Buffer Set, Biolegend, San Diego, CA, USA) and incubated with extracellular antibodies: anti-CD45, anti-CD206, anti-CD68, anti-CD11b, anti-CD4, and anti-CD3 (Biolegend, USA; Table 1) for 30 min at 4 °C, diluted in PBS-BSA-A (PBS—0.1% Bovine Serum Albumin—0.5% Sodium Azide). They were then washed with wash buffer (PBS-BSA-A) and fixed again with commercial buffers (kit True-Nuclear Transcription Buffer Set, Biolegend, USA).

Next, the cells were permeabilized with commercial buffer (kit True-Nuclear Transcription Buffer Set, Biolegend, San Diego, CA, USA) for 20 min and incubated with intracellular antibodies overnight: anti-GFAP, anti-TNF-α, anti-IFN-γ, anti-IL-10, and anti-IL-4 (Biolegend, USA; Table 1).

After the incubation period, the cells were washed with a wash buffer and fixed with commercial buffer (kit True-Nuclear Transcription Buffer Set, Biolegend, USA). The analyses were performed using a flow cytometer (NovoCyte—ACEA Biosciences, San Diego, CA, USA), and the data were analyzed using NovoExpress software (version 1.6.2), considering a minimum of 10,000 events per tube. The gating strategy used is described below (Figure 2), and it was applied to all the evaluated groups and the gating strategy used for the unstained control groups is provided in the Appendix A.

### 2.5. Statistical Analysis

For all groups evaluated in this study, the data were standardized as mean ± standard error of the mean (SEM). The results were subjected to one-way ANOVA, followed by Bonferroni post-test, with *p* < 0.05 considered as the significance threshold.

## 3. Results

The flow cytometry data obtained from the experimental groups were used for analyses involving subpopulations of astrocytes, microglia, macrophages, and lymphocytes after treatment with TEMPOL and DMF.

### 3.1. Astrocytes A1 and A2

Ventral root crush commonly leads to extensive local inflammation and the death of a large portion of axotomized motoneurons. Flow cytometry was performed on the ipsilateral side of the spinal cords, allowing the analysis of potential anti-inflammatory glial polarizations. Subpopulations of astrocytes were analyzed based on GFAP+ labeling and their respective pro- or anti-inflammatory cytokines. A1 astrocytes were characterized by a GFAP + TNF-α+ and GFAP + IFN-γ phenotype (Figure 3); A2 astrocytes, on the other hand, were characterized by a GFAP + IL-10+ and GFAP + IL-4+ phenotype (Figure 4).

According to the data herein, 7 days after injury, there was a decrease in TNF-α-producing A1 astrocytes in the TEMPOL experiment (TE) compared to the vehicle experiment (VE), with rates of 68% in the VE compared to 34% in the TE (VE: 68.16 ± 5.75 vs. TE: 34.52 ± 5.8, *p* = 0.0044). The DMF experiment (DMFE) showed the highest percentage of A1 astrocytes, with 74% compared to 34% in the TE (DMFE: 74.79 ± 5.83 vs. TE: 34.52 ± 5.80, *p* = 0.0011) (Figure 3A,D). After 14 days of treatment, there was also a decrease in the TE, from 55% to 9% (VE: 55.83 ± 8.45 vs. TE: 9.33 ± 2.83, *p* = 0.0003). At 14 dpi, the DMF group also presented the highest percentage (DMFE: 65.35 ± 4.53 vs. TE: 9.33 ± 2.83, *p* ≤ 0.00001) (Figure 3B,D). And at 28 dpi, the TE had the highest percentage of such A1 astrocytes, with 67% vs. 29% in the DMFE (TE: 67.6 ± 6.42 vs. 29.49 ± 8.63, *p* = 0.0051) (Figure 3C,D).

Regarding IFN-γ-producing astrocytes, 14 days after the injury, there was also a decrease in the TE, about 71% in the vehicle group *versus* 52% in the TEMPOL group (71.84 ± 7.27 vs. 52.06 ± 3.03, *p* = 0.0318). The DMF group compared to TEMPOL demonstrated that TE decreased A1 profile (DMFE: 85.43 ± 1.48 vs. TE: 52.06 ± 3.03, *p* = 0.0008) (Figure 3B,D). Thus, a greater anti-inflammatory effect was observed at 14 days of treatment with TEMPOL and at 28 days with DMF.

Regarding A2 astrocytes in the TEMPOL treatment, based on IL-10 and IL-4 labeling at 7, 14, and 28 days, no significant changes were observed (Figure 4). However, in the DMF treatment at 28 dpi, the TE group showed a higher percentage of IL-4-producing A2 astrocytes, while the DMFE group showed a lower percentage (VE: 8.65 ± 0.82 vs. DMFE: 1.59 ± 0.63, *p* = 0.0053; TE: 10.51 ± 1.90 vs. DMFE: 1.59 ± 0.62, *p* = 0.0008) (Figure 4A,B).

The decrease in A1 astrocytes and the lack of increase in A2 astrocytes in the TEMPOL group compared to the vehicle group suggest that there was likely an increase in the number of non-polarized astrocytes, meaning astrocytes that do not express a specific A1 or A2 phenotype (Figure 5).

### 3.2. Microglia M1 and M2

The analysis of microglia subpopulations was performed using CD11b*high* and CD45*low* labeling, along with their respective pro- or anti-inflammatory cytokines. M1 microglia were characterized by a CD11b + CD68 + TNF-α+ phenotype, while M2 microglia exhibited a CD11b + CD68 + IL-10+ phenotype (Figure 6).

When analyzing M1 microglia at 14 days of treatment, the rate decreased from 82% in the vehicle group to 30% in the Tempol group (VE: 82.36 ± 3.65 vs. TE: 30.17 ± 5.67, *p* = 0.0129) (Figure 6B,D). However, 28 days postinjury, there was an increase in M1 microglia in the TE compared to the VE, with approximately 43% in the vehicle group and 90% in the Tempol group (VE: 43.65 ± 14.99 vs. TE: 90.12% ± 2.08, *p* = 0.0284) (Figure 6C,D).

Observing M2 microglia, in both the TEMPOL and DMF-treated groups, there was a significant reduction at the 14 dpi. The vehicle group showed a decrease from 9% to 2% in the TEMPOL group (VE: 9.12 ± 2.92 vs. TE: 2.17 ± 0.81, *p* = 0.0473) (Figure 6B,D). The reduction from the vehicle to the DMF group was from 9% to 0.14% (VE: 9.12 ± 2.92 vs. DMFE: 0.14 ± 0.09, *p* = 0.0104) (Figure 6B,D).

Similar to astrocytes, TEMPOL treatment increased the presence of unpolarized microglia, particularly at 14 dpi, contributing to the prevalence of an anti-inflammatory phenotype (Figure 7).

### 3.3. Macrophages M1 and M2

The macrophage subpopulations were observed using CD11b + CD45+ labeling and their respective pro- or anti-inflammatory cytokines. M1 macrophages were identified by the phenotype CD11b + CD45 + CD206 + TNF-α+, while M2 macrophages were identified by the phenotype CD11b + CD45 + CD206 + IL-10+ (Figure 8).

No statistical differences were obtained at 7 and 28 dpi. However, at 14 dpi, the TEMPOL group demonstrated a significant decrease in M1 macrophages compared to the vehicle group, with a reduction from 70% in the VE to 23% in the TE (VE: 70.45 ± 6.2 vs. TE: 23.73 ± 4.1, *p* = 0.001) (Figure 8B,D). In contrast, between the TEMPOL and DMF groups, animals treated with DMF showed an increase in M1 macrophages at 14 days (TE: 23.73 ± 4.1 vs. DMFE: 50.2 ± 8.79, *p* = 0.0469) (Figure 8B,D). Interestingly, both TEMPOL and DMF administration result in concurrent downregulation of M2 macrophages at 14 dpi (VE: 38.21 ± 5.80 vs. TE: 20.05 ± 3.1, *p* = 0.0487), returning to basal levels at 28 dpi.

### 3.4. Lymphocytes Th1 and Th2

Regarding the lymphocyte subpopulations, they were labeled with CD3 + CD4+ and their respective pro- or anti-inflammatory cytokines. Th1 lymphocytes were characterized by a CD3 + CD4 + IFN-γ+ phenotype, while Th2 lymphocytes were characterized by CD3 + CD4 + IL-4+ (Figure 9).

Analyzing the data on Th1 lymphocytes, a decrease in these cells was observed at 7 days, with rates of 62% in the vehicle group compared to 31% in the group treated with TEMPOL (VE: 62.84 ± 9.72 vs. TE: 31.04 ± 7.38, *p* = 0.0372) (Figure 9A,D), and also at 14 days (VE: 48.05 ± 3.3 vs. TE: 17.13 ± 5.61, *p* = 0.0033) (Figure 9B,D). At this time point, among the treated groups, the DMF group showed the highest rate of Th1 lymphocytes (DMFE: 64.5 ± 6.02 vs. TE: 17.13 ± 5.61, *p* < 0.00001) (Figure 9B,D). At 28 dpi, the DMFE showed a reduction compared to the VE (VE: 47.58 ± 4.79 vs. DMFE: 21.9 ± 3.83, *p* = 0.0105); it also presented a lower rate when compared to the TE (TE: 56.52 ± 6.14 vs. DMFE: 21.9 ± 6.83, *p* = 0.0011) (Figure 9C,D).

Regarding Th2 lymphocytes, there was a decrease in the TE compared to the VE at 14 days, around 10% in the vehicle group and 2% in the group treated with TEMPOL (VE: 10.46 ± 1.33 vs. TE: 2.41 ± 0.71, *p* = 0.0011) (Figure 9B,D). Also, at this time point, among the treated groups, the DMFE showed the highest percentage (TE: 2.41 ± 0.71 vs. DMFE: 10.06 ± 1.34, *p* = 0.0017) (Figure 9B,D).

All relevant data from each cellular group have been systematically compiled and presented in the table below (Table 2).

## 4. Discussion

Proximal spinal cord injuries have become increasingly common in recent years due to the rising incidence of high-intensity trauma. As a result, there is a growing demand for pharmacological strategies capable of providing neuroprotection during the acute post-injury phase. In this context, our group has been investigating various molecules with immunomodulatory and neuroprotective properties to reduce the loss of spinal motoneurons during the critical post-injury period [23,24,25].

Immunomodulation is essential in neuroprotection and preserving the integrity of synaptic circuits in the spinal cord. The reduction of astroglial and microglial reactions, can promote the survival of axotomized neurons by lowering the levels of pro-inflammatory cytokines, reactive oxygen species (ROS), and nitric oxide (NOS) [26]. However, these processes also heighten cellular stress, contributing to the development of secondary injuries [27].

Pharmacological treatment has advantages, such as easy administration and continuous use, with a low risk of side effects [28]. In the present study, we tested two drugs with potential immunomodulatory and neuroprotective effects, TEMPOL (50 mg/kg) and DMF (15 mg/kg), which were administered daily during the treatment period after injury.

It was observed that motoneuron degeneration occurs at levels similar to the control group, suggesting the lack of direct neuroprotective effects with TEMPOL treatment. However, TEMPOL was associated with a reduction in the production of IFN-γ and TNF-α, highlighting its potential immunomodulatory properties [29]. These cytokines, TNF-α and IFN-γ, trigger a Th1-type inflammatory immune response, leading to the generation of reactive oxygen species (ROS) and reactive nitrogen species (RNS), cytotoxicity, and tissue damage [30]. Based on this, the administration of the higher dose of Tempol showed effectiveness in reducing these indicators of cellular stress. This resulted in the attenuation of microglial reactions, thus contributing to improved synaptic stability in the injured segments of the spinal cord [2,31].

TEMPOL reduces M1 but does not increase M2, suggesting a higher contingent of non-polarized microglia. This was also observed in an amyotrophic lateral sclerosis model using SOD1G93A mice treated with IFN-β, where administration resulted in 60% of cells polarizing to M1 or M2, leaving a significant contingent of cells unpolarized. [32].

Microglial reactions are an important aspect of spinal cord injuries. Microglial cells respond quickly to injury and remain active during the acute post-injury phase, which can last from 1 to 2 weeks [33]. Microglia play a role in synapse elimination, a process that is regulated by the reduction of pro-inflammatory events [34]. Furthermore, the limitation of microglial cell activation within the central nervous system (CNS) has been demonstrated by low MHC II expression in these cells, indicating that TEMPOL restricts the activation of the “first line” of the CNS immune response [35].

The same has been observed in macrophages, having a similar effect to microglia, as M1 macrophages/microglia can play a role in secondary tissue damage and axonal retraction that occurs after spinal cord injuries, while M2 polarization can be associated with protective effects and promotion of axonal growth observed in spinal cord and optic nerve injury models [36,37]. Complementarily, the reduction in the production of pro-inflammatory cytokines suggests a shift from a Th1-type of pro-inflammatory response to a more immunosuppressive Th2-type response. This shift has long been suggested to be beneficial in CNS degenerative disorders, representing the functional mechanism of immunomodulatory therapy by TEMPOL [29].

The neuroprotective role of DMF has been widely reported in scientific literature. It is well-established that the drug shields neural cells from oxidative damage by activating the Nrf2-ERK1/2 MAPK signaling pathway, which leads to elevated levels of antioxidant proteins, including HO-1, glutathione-S-transferase, superoxide dismutase, and quinone reductase-1 across different cell types [38,39,40,41]. Normally, DMF is used in the treatment of multiple sclerosis. Therefore, experimental studies that used this drug for treatment have shown a reduction in disease severity, especially concerning the drug’s neuroprotective mechanism [17,40].

Previous studies on crush and avulsion injuries have shown that treatment with DMF exhibited neuroprotective and immunomodulatory potential, particularly after a 4-week period [1,42]. Additionally, DMF inhibits aerobic glycolysis by modifying the enzyme glyceraldehyde-3-phosphate dehydrogenase (GAPDH), resulting in anti-inflammatory effects in macrophages and Th cells [43].

Our results showed that DMF treatment significantly reduces glial reactivity, as evidenced by the decrease in pro-inflammatory cytokines marking astrocytes and microglia, suggesting a potential immunomodulatory activity starting from the third week of treatment. Glial reactivity may be associated with synapse loss and neuronal death [44,45]. In a previous study, DMF treatment was able to reduce both microglial and astrocyte responses, demonstrating a long-lasting effect that persisted for several weeks after drug treatment [1,34]. Future experiments will address sex differences, as the glial response and speed of polarization may be different in males and females. Such investigations will be part of efforts to establish the Brazilian Center for Gender Biology and Medicine (CGBM) as part of a global collaboration initiated at the University of Saarland, Germany.

Axon and myelin fragmentation typically occur within a few days after a nerve injury; however, the complete removal of debris is generally finished around 14 days after the event [46,47]. Macrophages are crucial for the removal and remodeling of these tissue fragments, which explains the acute increase in their presence within the injured nerve at the 14-day [11,48]. It has been shown that DMF affects the balance between M1 and M2 macrophages [49], but our results did not show significant differences in the quantity of these cellular subpopulations between the DMF-treated group and the controls. Additionally, a trend toward a decrease in the M1 subpopulation was observed from 7 to 14 days in the DMF group. Furthermore, the profile of M2 macrophages was similar between the DMF-treated mice and controls, with only a slight, non-significant increase in these cells observed at days 7 and 28.

T lymphocytes are among the last immune cells to infiltrate the nerve during Wallerian degeneration, with their presence first detectable at the injury site on day 3 and peaking around 21 days after chronic sciatic nerve compression in rats [46,50,51]. Although previous studies have noted a rise in pro-inflammatory molecules during the acute phase of Wallerian degeneration, followed by anti-inflammatory ones [47], the exact dynamics of Th cell subpopulations during this process remain unclear. Given that DMF has been shown to reduce Th1 cell frequency while increasing Th2 lymphocyte frequency in multiple sclerosis patients [52]. This shift likely contributed to a more favorable environment for axonal regeneration in the periphery, resulting in functional recovery.

## 5. Conclusions

Overall, TEMPOL and DMF reduced pro-inflammatory cytokines but did not show significant results in increasing anti-inflammatory phenotypes. It was observed that they modulated the inflammatory response towards a more pro-regenerative profile, with TEMPOL having a greater effect in the first two weeks and DMF showing a more significant effect from the third week.

## Figures and Tables

**Figure 1 biology-14-00473-f001:**
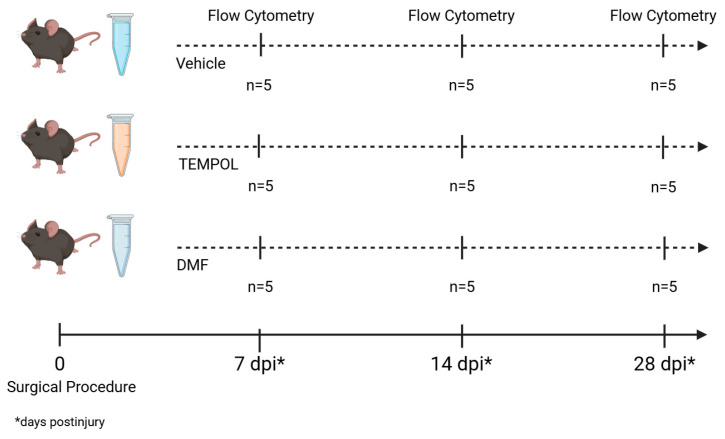
Experimental design. The three groups underwent a surgical procedure and were divided according to their treatment: vehicle, TEMPOL, and DMF. Flow cytometry analysis was performed at three distinct time points: 7, 14, and 28 days post-injury (dpi), with *n* = 5 at all time points.

**Figure 2 biology-14-00473-f002:**
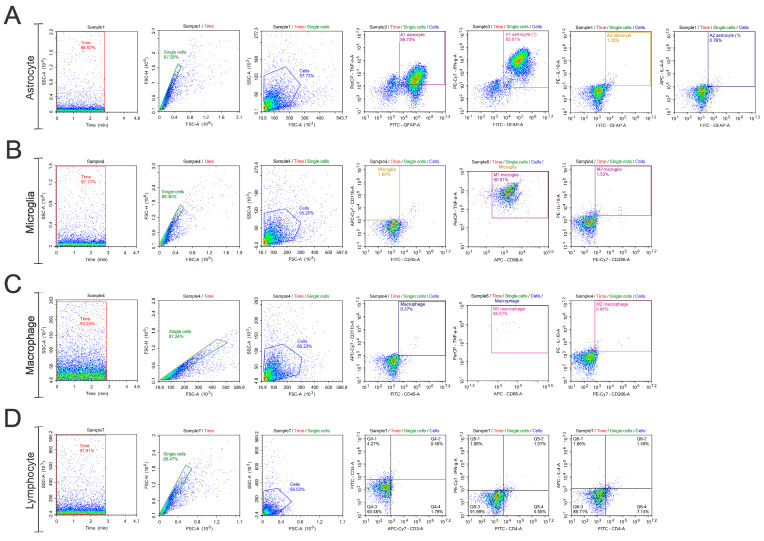
Gate strategy. Time gate initiation commenced the analysis, with doublets removed during single cell gating. To examine cell phenotype, a control gate using unlabeled cells was established. Doublets were eliminated by differentiating events using FSC-A and FSC-H parameters. A two-dimensional plot of FSC-A versus FSC-H visualized the doublet-free events, defining the “single cells” population. Following this, a dot plot combining complexity and size (FSC-A versus SCA-A) was created to determine the cell population. For assessing each marker fluorescence, a quadrant was placed on the edge of the population events, ensuring all subsequent events showed positive marker signals. This method was applied at each layer interface: (**A**) Astrocyte; (**B**) Microglia; (**C**) Macrophage; (**D**) Lymphocyte.

**Figure 3 biology-14-00473-f003:**
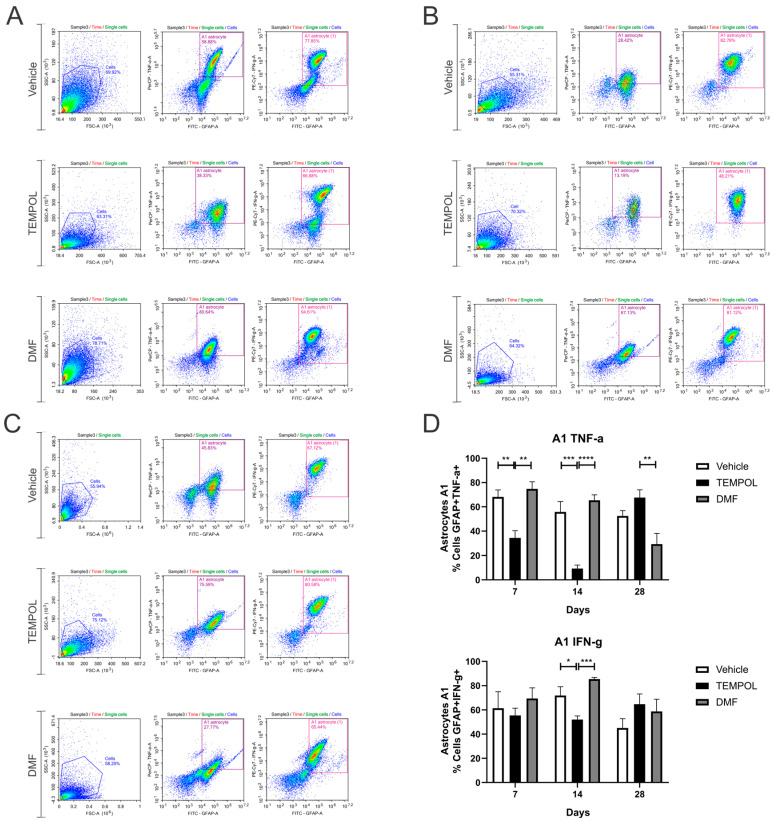
Representative dot plots of A1 astrocyte characterization, with the first analysis in FSC (size) by SSC (complexity), selecting the gate for subsequent analysis of GFAP + TNF-α+ and GFAP + IFN-γ+ astrocyte phenotypes: (**A**) Vehicle, TEMPOL and DMF groups at 7 dpi; (**B**) Vehicle, TEMPOL and DMF groups at 14 dpi; (**C**) Vehicle, TEMPOL and DMF groups at 28 dpi; (**D**) Comparison of the percentage of labeled cells at treatment times of 7, 14, and 28 days. Two-way ANOVA, *p* < 0.05 (*), *p* < 0.001 (**), *p* < 0.0001 (***), *p* < 0.00001 (****).

**Figure 4 biology-14-00473-f004:**
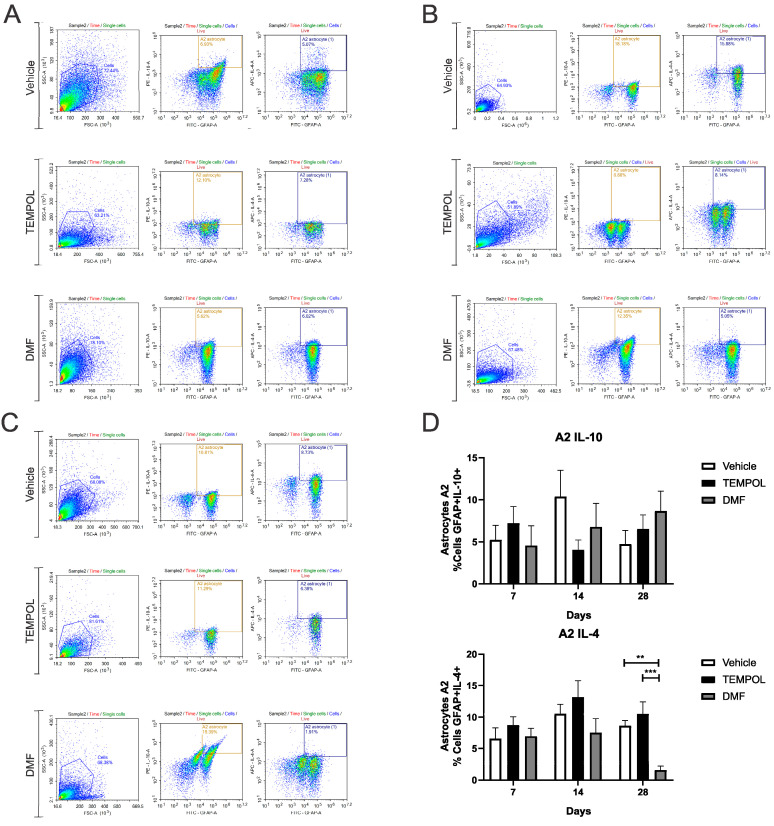
Representative dot plots of A2 astrocyte characterization, with the first analysis in FSC (size) by SSC (complexity), selecting the gate for subsequent analysis of GFAP + IL-10+ and GFAP + IL-4+ astrocyte phenotypes: (**A**) Vehicle, TEMPOL and DMF groups at 7 dpi; (**B**) Vehicle, TEMPOL and DMF groups at 14 dpi; (**C**) Vehicle, TEMPOL and DMF groups at 28 dpi; (**D**) Comparison of the percentage of labeled cells at treatment times of 7, 14, and 28 days. Two-way ANOVA, *p* < 0.001 (**), *p* < 0.0001 (***).

**Figure 5 biology-14-00473-f005:**
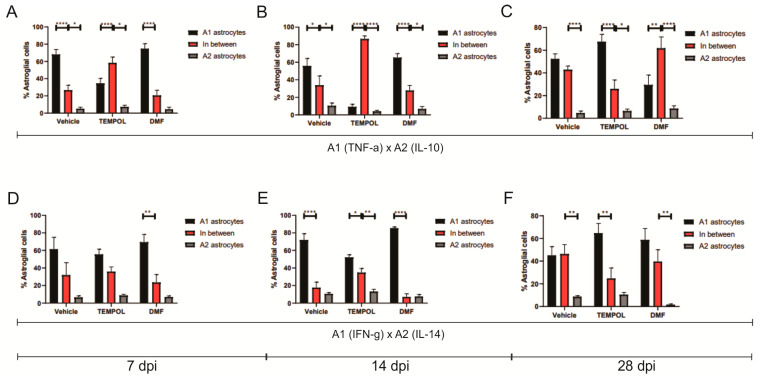
Representative graphs of the total percentage of A1, intermediate, and A2 astrocytes in the vehicle, TEMPOL and DMF groups: (**A**) 7 dpi: A1 (TNF-α) and A2 (IL-10); (**B**) 14 dpi: A1 (TNF-α) and A2 (IL-10); (**C**) A1 (TNF-α) and A2 (IL-10); (**D**) 7 dpi; A1 (IFN-γ) and A2 (IL-4); (**E**) A1 (IFN-γ) and A2 (IL-4); (**F**) A1 (IFN-γ) and A2 (IL-4). Two-way ANOVA, *p* < 0.05 (*), *p* < 0.001 (**). *p* < 0.00001 (****).

**Figure 6 biology-14-00473-f006:**
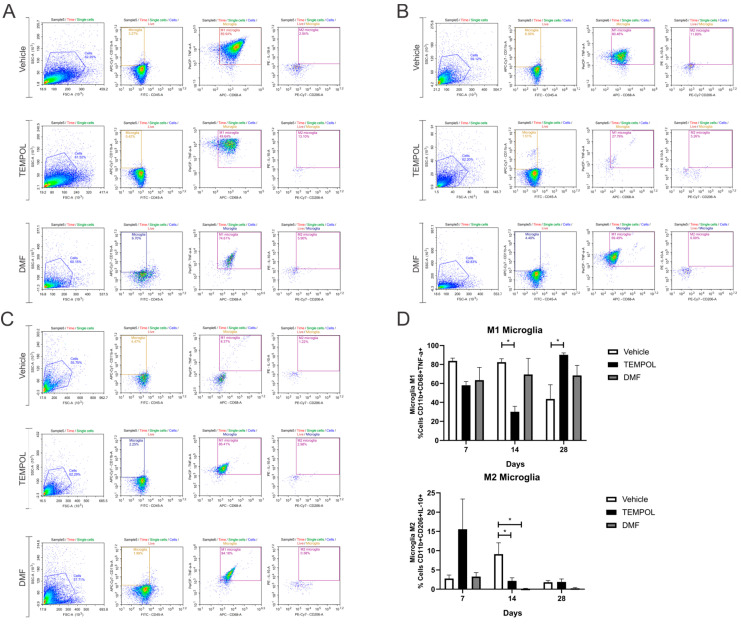
Representative dot plots for the characterization of M1 and M2 microglia, with the first analysis based on FSC (size) and SSC (complexity), followed by gate selection for the subsequent analysis of microglia phenotypes CD11b + CD68 + TNF-α+ and CD11b + CD68 + IL-10+, respectively (**A**) Vehicle, TEMPOL and DMF groups at 7 dpi; (**B**) Vehicle, TEMPOL and DMF groups at 14 dpi; (**C**) Vehicle, TEMPOL and DMF groups at 28 dpi; (**D**) Comparison of the percentage of labeled cells at treatment times of 7, 14, and 28 days. Two-way ANOVA, *p* < 0.05 (*).

**Figure 7 biology-14-00473-f007:**
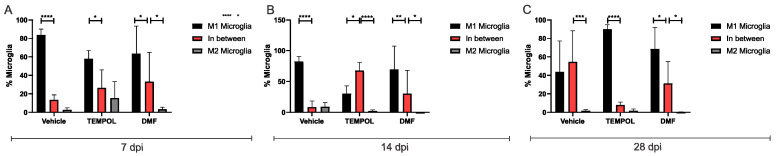
Representative graphs of the total percentage of M1, intermediate, and M2 microglia in the vehicle, TEMPOL and DMF groups: (**A**) 7 dpi; (**B**) 14 dpi; (**C**) 28 dpi. Two-way ANOVA, *p* < 0.05 (*), *p* < 0.001 (**), *p* < 0.0001 (***), *p* < 0.00001 (****).

**Figure 8 biology-14-00473-f008:**
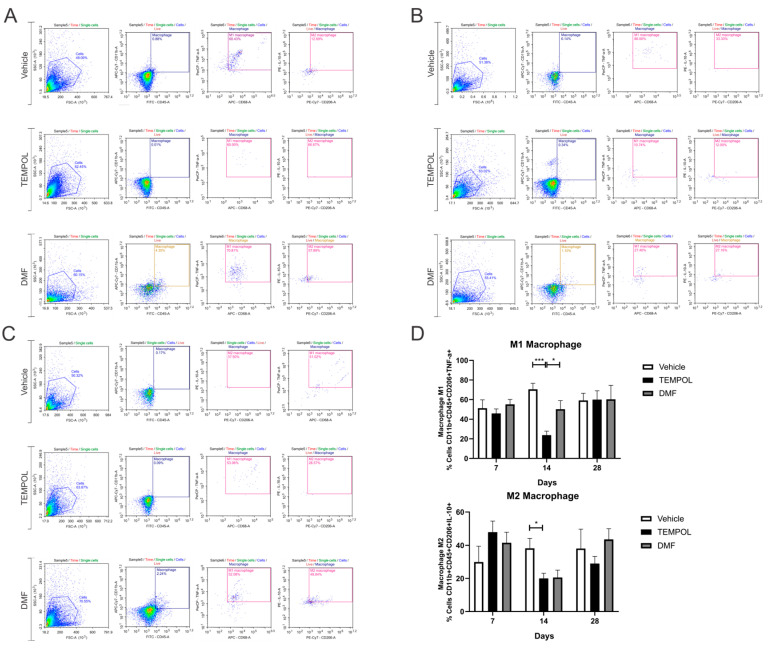
Representative dot plots for the characterization of M1 and M2 macrophages, with the first analysis in FSC (size) by SSC (complexity), followed by gate selection for further analysis of macrophages with the phenotypes CD11b + CD45 + CD206 + TNF-α+ and CD11b + CD45 + CD206 + IL-10+, respectively: (**A**) Vehicle and Tempol group at 7 dpi; (**B**) Vehicle and Tempol group at 14 dpi; (**C**) Vehicle and Tempol group at 28 dpi; (**D**) Comparative analysis of the percentage of labeled cells at 7, 14, and 28 days of treatment. Two-way ANOVA, *p* < 0.05 (*), *p* < 0.0001 (***).

**Figure 9 biology-14-00473-f009:**
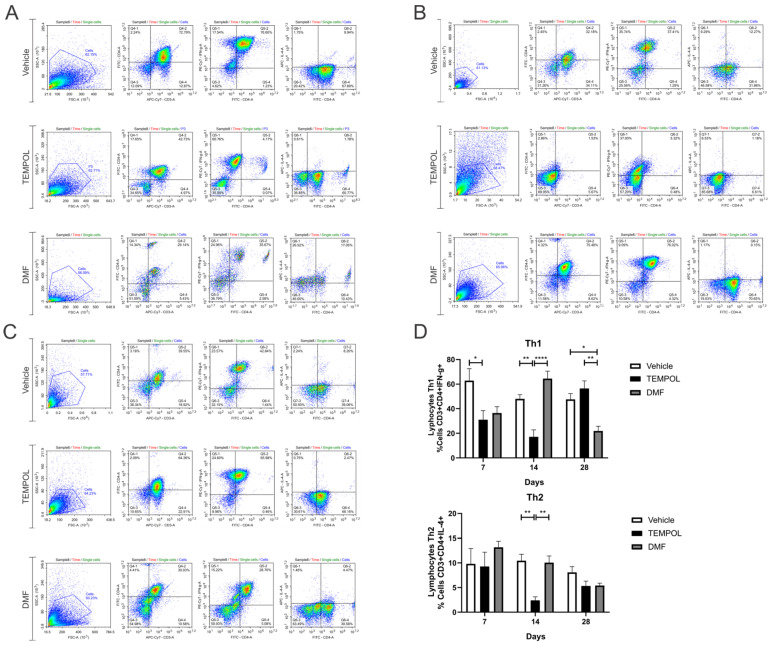
Representative dot plots of Th1 and Th2 lymphocyte characterization, with the initial analysis in FSC (size) by SSC (complexity), selecting the gate for subsequent analysis of Th1 lymphocytes with the phenotype CD3 + CD4 + IFN-γ+ and Th2 lymphocytes with CD3 + CD4 + IFN-γ+: (**A**) Vehicle and Tempol group at 7 dpi; (**B**) Vehicle and Tempol group at 14 dpi; (**C**) Vehicle and Tempol group at 28 dpi; (**D**) Comparative analysis of the percentage of labeled cells across the treatment periods of 7, 14, and 28 days. Two-way ANOVA, *p* < 0.05 (*), *p* < 0.001 (**), *p* < 0.00001 (****).

**Table 1 biology-14-00473-t001:** Antibodies used for flow cytometry.

TUBE	FITC	PE	PerCP	PE-Cy7	APC	APC-Cy7
1	-	-	-	-	-	-
2	GFAP	IL-10			IL-4	
3	GFAP		TNF-α	IFN-γ		
4	-	-	-	-	-	-
5	CD45	IL-10		CD206		CD11b
6	CD45		TNF-α		CD68	CD11b
7	-	-	-	-	-	-
8	CD4			IFN-γ	IL-4	CD3

**Table 2 biology-14-00473-t002:** Statistical analysis of the main data obtained from astrocytes, microglia, macrophages, and lymphocytes following treatment with the vehicle, TEMPOL, and DMF after 7, 14, and 28 days (dpi). Two-way ANOVA, *p* < 0.05 (*), *p* < 0.001 (**), *p* < 0.0001 (***), *p* < 0.00001 (****).

Analysis	Time	Comparation	Statistical Data	*p* Value	*
Flow Cytometry					
**A1 Astrocyte** (**TNF-α^+^)**	7 dpi	VE vs. TE	68.16 ± 5.75 vs. 34.52 ± 5.80	0.0044	**
	7 dpi	DMFE vs. TE	74.79 ± 5.83 vs. 34.52 ± 5.80	0.011	**
	14 dpi	VE vs. TE	55.83 ± 8.45 vs. 9.33 ± 2.83	0.0003	***
	14 dpi	DMFE vs. TE	65.35 ± 4.53 vs. 9.33 ± 2.83	<0.00001	****
	28 dpi	TE vs. DMFE	67.60 ± 6.42 vs. 29.49 ± 8.63	0.0051	**
**A1 Astrocyte (IFN-γ^+^)**	14 dpi	VE vs. TE	71.84 ± 7.27 vs. 52.06 ± 3.03	0.0318	*
	14 dpi	DMFE vs. TE	85.43 ± 1.48 vs. 52.06 ± 3.03	0.0008	***
**A2 Astrocyte (IL-4)**	28 dpi	VE vs. DMFE	8.65 ± 0.82 vs. 1.59 ± 0.63	0.0053	**
	28 dpi	TE vs. DMFE	10.51 ± 1.90 vs. 1.59 ± 0.62	0.0008	***
**M1 Microglia**	14 dpi	VE vs. TE	82.36 ± 3.65 vs. 30.17 ± 5.67	0.0129	*
	28 dpi	VE vs. TE	43.65 ± 14.99 vs. 90.12 ± 2.08	0.0284	*
**M2 Microglia**	14 dpi	VE vs. TE	9.12 ± 2.92 vs. 2.17 ± 0.81	0.0473	*
	14 dpi	VE vs. DMFE	9.12 ± 2.92 vs. 0.14 ± 0.09	0.0104	*
**M1 Macrophage**	14 dpi	VE vs. TE	70.45 ± 6.20 vs. 23.73 ± 4.10	0.001	***
	14 dpi	TE vs. DMFE	23.73 ± 4.10 vs. 50.2 ± 8.79	0.0469	*
**M2 Macrophage**	14 dpi	VE vs. TE	38.21 ± 5.80 vs. 20.05 ± 3.10	0.0487	*
**Th1 Lymphocyte**	7 dpi	VE vs. TE.	62.84 ± 9.72 vs. 31.04 ± 7.38	0.0372	*
	14 dpi	VE vs. TE.	48.05 ± 3.30 vs. 17.13 ± 5.61	0.0033	**
	14 dpi	DMFE vs. TE	64.50 ± 6.02 vs. 17.13 ± 5.61	<0.00001	****
	28 dpi	VE vs. DMFE	47.58 ± 4.79 vs. 21.9 ± 3.83	0.0105	*
	28 dpi	TE vs. DMFE	56.52 ± 6.14 vs. 21.90 ± 6.83	0.0011	**
**Th2 Lymphocyte**	14	VE vs. TE	10.46 ± 1.33 vs. 2.41 ± 0.71	0.0011	**
	14	TE vs. DMFE	2.41 ± 0.71 vs. 10.06 ± 1.34	0.0017	**

## Data Availability

The original contributions presented in this study are included in the article/Appendix A. Further inquiries can be directed to the corresponding author(s).

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
