# Peer review of "Immunomodulation by 4-Hydroxy-TEMPO (TEMPOL) and Dimethyl Fumarate (DMF) After Ventral Root Crush (VRC) in C57BL/6J Mice: A Flow Cytometry Analysis"

_biology, 2025, doi:10.3390/biology14050473_

Round 1

Reviewer 1 Report

Comments and Suggestions for Authors

Comments and Suggestions for Authors

This is an interesting manuscript dealing with the attempt to immunomodulation by 4-hydroxy-tempo and dimethyl fumarate after ventral root crush in C57BL/6J mice. In general, the study is well conducted.

There are three specific concerns: 

Introduction - more attention should be given to prior studies. Authors can find more recent studies which are used TEMPOL like antioxidants with other animal models. Referenceses can be updated in this manner.

Figure 1 and 3 have insuficiently described in the legend. The description should be sufficient that readers who have not conducted such experiments understand what they depict. 

General advice

Although the language of the study is understandable and easy-to-read, there are many outdated references. Supporting it with fewer and more up-to-date references will be more critical for the reader to keep up with the current situation on this study topic.

Author Response

  1. Introduction - more attention should be given to prior studies. Authors can find more recent studies which are used TEMPOL like antioxidants with other animal models. References can be updated in this manner.

AUTHOR RESPONSE & ACTION TAKEN: Recent studies using TEMPOL have been added to the manuscript. Articles were highlighted (lines 85-87) in the manuscript.

  1. Figure 1 and 3 have insuficiently described in the legend. The description should be sufficient that readers who have not conducted such experiments understand what they depict.

AUTHOR RESPONSE & ACTION TAKEN: The legends have been modified and detailed. Legends were highlighted (lines 125-128) in the manuscript.

Reviewer 2 Report

Comments and Suggestions for Authors

Thank you very much for the opportunity to review this article about Immunomodulation by 4-hydroxy-tempo (TEMPOL) and dimethyl fumarate (DMF) after ventral root crush (VRC) in C57BL/6J mice: a flow cytometry analysis. Overall I found it to be a bold article that offers a different approach and, therefore, may offer a valuable perspective.

However, some minor aspects are amenable to revision. These are as follows:

In the Methods section, I propose that at the beginning, they describe in a general way the type of study being performed. In fact, When was the study done?

In the Results section, it would be attractive to collect the data obtained in a table in a synthetised way.

Please, format and standarize the references (edit et al., just to add this after 6 authors).

Author Response

  1. In the Methods section, I propose that at the beginning, they describe in a general way the type of study being performed. In fact, When was the study done?

AUTHOR RESPONSE & ACTION TAKEN: At the beginning of the methods section, a brief summary of the type of study conducted was added. This new part has been highlighted (lines 111-114).

  1. In the Results section, it would be attractive to collect the data obtained in a table in a synthetised way.

AUTHOR RESPONSE & ACTION TAKEN: At the end of the results section, a table was added with the summarized data. The table has been highlighted (lines 332-334) in the manuscript.

  1. Please, format and standardize the references (edit et al., just to add this after 6 authors).

AUTHOR RESPONSE & ACTION TAKEN: The references have been properly formatted and updated in the manuscript based on the highlighted section at line 445.

Reviewer 3 Report

Comments and Suggestions for Authors
  1. The authors performed fixation and intracellular staining for markers such as GFAP and CD11b. While intracellular cytokine staining requires fixation, the assessment of surface markers such as CD11B, CD45, CD206 could have been performed on live cells. Fixing cells might introduce artifacts. The authors should justify why fixation was necessary for all flow cytometry markers.
  2. The classification of microglia into M1 and M2 subtypes is outdated. Current research recognizes that microglia exhibit a spectrum of activation states rather than binary polarization. The authors should update their terminology, if possible to include markers that better define microglial heterogeneity, such as Tmem119 for homeostatic microglia or Clec7a/SPP1 for disease-associated microglia. At the very least, a discussion on the limitations of the M1/M2 model should be included.
  3. P values are missing in Figures 6-8.
  4. Representative plots of negative controls, single antibody stained controls should be provided in the supplement. 

Author Response

  1. The authors performed fixation and intracellular staining for markers such as GFAP and CD11b. While intracellular cytokine staining requires fixation, the assessment of surface markers such as CD11B, CD45, CD206 could have been performed on live cells. Fixing cells might introduce artifacts. The authors should justify why fixation was necessary for all flow cytometry markers.

AUTHOR RESPONSE & ACTION TAKEN: Due to the multiple antibody staining, this step was essential to ensure cell preservation and accessibility of intracellular antigens, as the analysis performed was comprehensive of different cell types in a single sample, necessitating a standardized methodology for combined analysis with multiple markers.

  1. The classification of microglia into M1 and M2 subtypes is outdated. Current research recognizes that microglia exhibit a spectrum of activation states rather than binary polarization. The authors should update their terminology, if possible to include markers that better define microglial heterogeneity, such as Tmem119 for homeostatic microglia or Clec7a/SPP1 for disease-associated microglia. At the very least, a discussion on the limitations of the M1/M2 model should be included.

AUTHOR RESPONSE & ACTION TAKEN: Microglia exhibit a spectrum of activation with broad functional complexity and intermediate states. However, in this study, we limited our approach to the M1 (pro-inflammatory) and M2 (anti-inflammatory) polarization model, as it allows for an initial characterization of the microglial response to ventral root injury, helping to distinguish between inflammatory and reparative phases. On the other hand, future approaches should consider a more diverse range of microglial states for a more comprehensive understanding of neuroinflammation and regeneration, incorporating additional markers and transcriptomic analyses. Therefore, since the objective of this project was to broadly and directly analyze cellular subpopulations, this model effectively fulfilled its purpose.

  1. P values are missing in Figures 6-8.

AUTHOR RESPONSE & ACTION TAKEN: The p-values have been added to the figures and are highlighted (lines 283, 304 and 331) in the manuscript.

  1. Representative plots of negative controls, single antibody stained controls should be provided in the supplement.

AUTHOR RESPONSE & ACTION TAKEN: The supplementary material has been prepared and is attached.

Round 2

Reviewer 1 Report

Comments and Suggestions for Authors

Manuscript is accepted in present form.

Reviewer 2 Report

Comments and Suggestions for Authors

Overall Recommendation: Accept in present form

Reviewer 3 Report

Comments and Suggestions for Authors

The authors have addressed the reviewer's comments appropriately.